# Exploring Model-Based Shielding for Non-Player Character Behaviour

**William Ahlberg**[1,2]**, Konrad Tollmar**[2]**, Alessandro Sestini**[2]**, Linus Gisslén**[2]**, Iolanda Leite**[1]

`{wiahlb,iolanda}@kth.se,`
`{wahlberg,ktollmar,asestini,lgisslen}@ea.com`

[1]**Division of Robotics, Perception and Learning, KTH Royal Institute of Technology, Sweden**
[2]**SEED - Electronic Arts (EA)**

## Abstract

Reinforcement Learning can create agents that are able to play games at a human, or even super-human, level. Shielding in Reinforcement Learning is a technique used in robotics to enforce safe decision-making during both learning and execution, and allows robots to perform tasks safely. We explored how shields can be repurposed to align an agent to follow a designed *style specification* for more human-like and believable Non-Player Character behaviour in video games. Shielding can alleviate the need for extensive reward shaping when designing qualitative behaviours. However, classical shielding is often too restrictive in its assumptions to be easily applied to complex 3D environments with large continuous state spaces, such as video games. We proposed the use and repurposing of *Approximate Model-based Shielding* (AMBS) for video game purposes. We explored how AMBS can be used to introduce a style aspect to a task policy.

## 1 Introduction

In modern video games, Non-Player Character (NPC) behaviour is commonly created using either scripting or classical AI techniques like finite-state machines, behaviour trees, and utility systems (Yannakakis & Togelius, 2018). Historically these methods have been successful in creating compelling NPCs that enhance the gameplay experience; however, the effectiveness of these methods experience diminishing returns for video games with increasing complexity and scale, such as in AAA games. The manual labour for designing NPC behaviours is both expensive and time-consuming (Ahlberg et al., 2023; Sestini et al., 2023).

Machine Learning (ML) has been shown to scale well with data and computation, and ML-based agents have demonstrated potential being applied to video games. Reinforcement Learning (RL) has especially been successful in generating complex policies able to play games at a human, or even super-human, level. RL agents have shown complex decision making in games like Starcraft 2, Dota 2 and Minecraft (Vinyals et al., 2019; Berner et al., 2019; Hafner et al., 2025). Even recently, Wurman et al. (2022) presented Sophy, a super-human driver agent which has been commercially deployed in the racing simulator game Gran Turismo 7. Still, the utility of RL can be limited due to the difficulty of accurately defining a reward function for complex and qualitative behaviours, in particular believability and human likeness (Zhao et al., 2020; Sestini et al., 2021; Roy et al., 2022). While it is possible to create agents able to play games well, this is not the sole goal for game developers, as such agents might show unnatural, unintended, and distracting behaviours which ultimately worsen the gameplay experience. It can require extensive reward engineering by those with in-depth domain knowledge to generate a satisfying reward function (Aytemiz et al., 2021).

In robotics, RL is not only used to teach robots to perform tasks, such as manipulation and navigation, but also how to do so in a safe manner for humans and their surroundings. There is a plethora of work in the robotics community on how learning-based decision making can be aligned follow human defined safety rules (García & Fernández, 2015; Alshiekh et al., 2018; Hewing et al., 2020). *Safe Reinforcement Learning via Shielding*, as proposed by Alshiekh et al. (2018), is a commonly used method to incorporate safety features into a task objective. A *shield* is a component separate from the policy that can be added to the RL training loop; the shield oversees the actions of an agent and intervenes if the shield deems the action to be unsafe. The action is substituted with a "safe" action by the shield. Shielding can also be beneficial for sample efficiency as it restricts the search space of the RL agent, preventing it from exploring or exploiting unsafe states and actions.

While safety is not a primary concern for an NPC in a video game, in this work we show how shielding might be used to steer an agent's behaviour towards more human-like and believable qualities. We explore how specifically *Approximate Model-Based Shielding* (AMBS), formulated by Goodall & Belardinelli (2023), can be used to control NPC behaviour with shields upholding *style specifications*.

## 2  Related Work

### 2.1  Model-based Reinforcement Learning

In Model-based Reinforcement Learning (MBRL), the agent is able to plan its actions by accessing a dynamics model. The model allows the agent to try different trajectories without updating the environment, making the method possibly more sample-efficient. This is in contrast with Model-free Reinforcement Learning where agent actions update with the environment, making its actions irreversible. The Dreamer models (Hafner et al., 2019) are Recurrent State-Space Models (RSSMs) that learn a low-dimensional latent-space representation of the environment to learn an RL policy. MBRL with Dreamer has shown state-of-the-art performance in a multitude of different domains, while also being more sample-efficient, and requires less hyperparameter-tuning compared to model-free methods (Hafner et al., 2025). *DayDreamer* (Wu et al., 2023) allowed a quadruped robot to learn within 10 minutes how to roll of its back and walk, while *DreamerV2* (Hafner et al., 2020) managed to learn an RL policy the Atari games only in the latent-space of the model, and with better sample-efficiency compared to model-free baselines.

### 2.2  Shielding for Reinforcement Learning

The seminal work by Alshiekh et al. (2018) proposed shielding for Reinforcement Learning (RL) as a reactive system that upholds a predefined safety specification for an agent policy during both training and execution. The shield monitors the actions of the agent and prevents unsafe behaviour by substituting the agent's action $a$ with a safe action $a_{\text{safe}}$. It is assumed that an abstraction of the safety-relevant dynamics of the environment is accessible for a model-checker to verify and label reachable state-action pairs as either safe or unsafe. In practice, it is not always feasible to require full knowledge of the environment dynamics, and for large continuous state-spaces it can be computationally hard to compute safety labels. Bounded Prescience Shielding (BPS) omits the need for full knowledge of the safety-relevant dynamics by instead assuming access to a forward-simulator of the environment, and only needs to know the dynamics up to a bound $H \in \mathbb{N}$ (Giacobbe et al., 2021). Latent shielding and Approximate Model-based Shielding (AMBS) forgo the assumption of a forward-simulator as it can either be too computationally expensive and slow, or not present at all, and instead tries to learn the environment dynamics and safety-relevant properties by constructing a compact latent representation of the environment with a learned world model (He et al., 2021; Goodall & Belardinelli, 2023). The agent can use the learned dynamics model to "imagine" and plan future trajectories in the latent space, using its task policy, and avoid unsafe states. While shielding has primarily been used for safety, here we propose that it can be repurposed to design NPC behaviour by defining shields that both monitor and correct agents for unwanted behaviour.

## 3   Method

In a game environment, the goal of the algorithm is to produce a shielded policy for the game-playing agent that both plays the game well while also adhering to a set of predefined style specifications. The user (e.g. a player or game designer) defines the specifications as shields, which encode sought after behaviours, and are separate from the environment reward. It instead includes a set of stylistic states. Shields are specified with boolean statements. The algorithm is thoroughly described in the following Sections 3.1 to 3.5.

### 3.1   Problem Statement

The environment is modelled as a *Partially Observable Markov Decision Process* (POMDP) with labels, and is defined with the tuple $M = (S, A, P, R, \gamma, \Omega, O, AP, L)$ (Baier & Katoen, 2008; Bouton et al., 2020). Here $S$ and $A$ are the state and action spaces, respectively; $P : S \times A \times S \to [0, 1]$ is the transition function, where $P(s_{t+1}|s_t, a_t)$ denotes the probability of transitioning to the next state $s_{t+1}$ by taking action $a_t$ at state $s_t$; $R : S \times A \to \mathbb{R}$ is the reward function; $\gamma \in (0, 1]$ is the discount factor; $\Omega$ is the set of observations with $O : S \times \Omega \to [0, 1]$ being the observation function. In addition to the ordinary definition of a POMDP, states are also annotated with labels $L : S \to 2^{AP}$ where $L$ is the labelling function. States can be labelled with *atomic propositions* from the set $q \in AP$. Each atomic proposition is a boolean statement which cannot be broken down into smaller statements. In a video game setting, statements such as $q_0 = $ `player-out-of-bounds` or $q_1 = $ `player-inventory-full` are atomic propositions and can be made into more complex specifications with logical operations like *and* and *or*.

The agent has two separate policies: the *task* policy $\pi_{\text{task}}$, and the *style* policy $\pi_{\text{style}}$. The goal of the $\pi_{\text{task}}$ policy is to maximise cumulative reward with the optimal task policy being $\pi_{\text{task}}^* = \arg\max_\pi \mathbb{E}\left[\sum_{t=1}^\infty \gamma^{t-1} \cdot r_t\right]$. In a game environment, we define the task policy to be concerned with *what* the agent should do to beat the game. For example, win, obtain high-scores, or complete levels as fast as possible. The equivalent task reward function can often be clearly defined: maximise win-rate or high-score, and minimise game time. In contrast, the style of an agent is related to *how* a task is performed. Different styles can equate to differences in preference for certain strategies, items, and movement paths. For example, if the behaviour of a driver agent is reckless, preferring high acceleration, tight turns and aggressive overtaking; or careful, smooth inputs and less acceleration. With conventional RL, we cannot guarantee the style of the optimal policy, only that it is reward maximising.

With shielding, the proposed style of an agent is encoded with a style specification $\Psi$. As an example, consider the propositions: $q_0 = $ `player-health` $> 50\%$ and $q_1 = $ `enemy-nearby`. A risk-adverse agent would always try to keep $q_0$ true as a safety precaution, therefore an apt style specification would be:

$$\Psi_{\text{risk-averse}} = \texttt{player-health} > 50\,\%, \tag{1}$$

which reads: never allow the player's health to go below $50\%$. A risk-willing agent might instead not have the same worry, and conserve resources, and only heal if necessary. The style specification for the risk-willing agent could instead be:

$$\Psi_{\text{risk-willing}} = \texttt{player-health} > 50\,\% \wedge \texttt{enemy-nearby}. \tag{2}$$

It should only take actions to restore health if an enemy is nearby. We consider a style-following policy to minimise the total number of specification violations, where each violation incur a cost. Hence, the optimal style policy is $\pi_{\text{style}}^* = \arg\min_{\pi_{\text{style}}} \mathbb{E}\left[\sum_{t=1}^\infty \gamma^{t-1} \cdot c_t\right]$. However, if maximising the reward and minimising the violation cost are conflicting goals, the policy would not converge. The optimal policy from a *Constrained Markov Decision Problem* (CMDP) perspective is defined as $\pi^* = \arg\max_{\pi \in \Pi_c} \mathbb{E}\left[\sum_{t=1}^\infty \gamma^{t-1} \cdot r_t\right]$ which is the optimal policy from the feasibility set $\Pi_c$. How to balance the goals between the task and style policies is further explained in Section 3.4.

## 3.2 Probabilistic Computation Tree Logic

To label states, style specifications are written using *Probabilistic Computation Tree Logic* (PCTL) defined by Hansson & Jonsson (1994). PCTL is a specification language for expressing real-time stochastic systems. Its syntax is defined as follows:

1. Each atomic proposition $q$ is a PCTL formula,

2. if $\phi_1$ and $\phi_2$ are PCTL formulas, then so are $\neg\phi_1$ and $\phi_1 \wedge \phi_2$,

3. if $\phi_1$ and $\phi_2$ are PCTL formulas, $t$ is a non-negative integer or $\infty$, and $p$ is a real number with $0 < p < 1$, then $\phi_1 U_{\geq p}^{\leq t}\phi_2$ and $\phi_1 U_{> p}^{\leq t}\phi_2$ are PCTL formulas.

The operators $\neg$ and $\wedge$ are the logical *not* and *and* operators, respectively. $U$ is a temporal operator called the *until* operator, where the formula $\phi_1 U_{\geq p}^{\leq t}\phi_2$ expresses that with at least the probability $p$, $\phi_1$ will be true for $t$ timesteps until $\phi_2$ becomes true after. To reason and express regarding sequences of states, a structure is defined with the quadruple tuple $S, s_0, P, L$ where:

1. $S$ is a set of ordered states $s_0, s_1, s_2, \ldots$,

2. $s_0$ is the *initial state*,

3. $P : S \times S \rightarrow [0, 1]$ is the transition probability function,

4. $L : S \rightarrow 2^{AP}$ is a labelling function assigning atomic propositions from the set $q \in AP$ to states.

The POMDP with labels facilitates the use of PCTL to express style specifications which are explained in Section 3.1. Violating this style specification will incur the agent a cost. It is up to the style policy $\pi_{\text{style}}$ to learn how to uphold the style specification for the agent.

## 3.3 Approximate Model-based Shielding

Approximate Model-based Shielding (AMBS) is a shielding framework proposed by Goodall & Belardinelli (2023) for safe RL. It uses a World Model to learn safety-relevant dynamics of the environment and an optimal RL policy network. Look-ahead shielding is performed within its latent-space, and does not require a known model of the environment which is unlike many other shielding methods Odriozola-Olalde et al. (2023). AMBS, and we, use *DreamerV3* as its stand-in World Model. DreamerV3 is a *Recurrent State Space Model* (RSSM) (Hafner et al., 2019) which is a type of sequential *Variational Autoencoder* (VAE). Since DreamerV3 efficiently uses a compact latent-space for both model and policy learning, it is proven to be well suited to work in continuous and high-dimensional state spaces. Like the original implementation of AMBS, we also use DreamerV3 (Hafner et al., 2025) as our World Model. The model components are defined as:

$$\text{Sequential model: } h_t = f_\theta(h_{t-1}, z_{t-1}, a_{t-1}),$$
$$\text{Observation encoder: } z_t \sim q_\theta(z_t|o_t, h_t),$$
$$\text{Transition predictor: } \hat{z}_t \sim p_\theta(\hat{z}_t|h_t),$$
$$\text{Observation decoder: } \hat{o}_t \sim p_\theta(\hat{o}_t|h_t z_t),$$
$$\text{Reward predictor: } \hat{r}_t \sim p_\theta(\hat{r}_t|h_t, z_t),$$
$$\text{Continuation predictor: } \hat{\gamma}_t \sim p_\theta(\hat{\gamma}|h_t, z_t),$$
$$\text{Cost predictor: } \hat{c}_t \sim p_\theta(\hat{c}_t|h_t, z_t),$$

where $h_t$ is the recurrent state conditioned on the previous recurrent state $h_{t-1}$, the stochastic latent $z_{t-1}$, and previous action $a_{t-1}$. The transition predictor predicts the prior $\hat{z}_t$ of $z_t$. The latent representation of the environment state is learnt by minimising the reconstruction loss between the observation $o_t$ and reconstructed observation $\hat{o}_t$. The model-heads for sampling $\hat{r}_t, \hat{\gamma}_t, \hat{c}_t, \hat{\gamma}_t^c$ are conditioned on the recurrent and latent state, and are used for policy learning of the task policy $\pi_{\text{task}}$ and style policy $\pi_{\text{style}}$.

### 3.4 Policy Learning

Both $\pi_{\text{task}}, \pi_{\text{style}}$ are trained using Actor-Critic learning as described by Hafner et al. (2025). The actor and critic neural networks are concurrently trained from a replay experience buffer, while the agent also performs environment interactions. Gradients are updated using the REINFORCE estimator (Williams, 1992) with a Penalty Critic (PENL) as proposed by Goodall & Belardinelli (2024). TD3 (Fujimoto et al., 2018) is used to combat overestimation, and two critic $v_1^C, v_2^C$ are learned where the minimum of the two is used. The PENL techniques balances task and style policy objectives as discussed in Section 3.1. From a given state, PENL estimates the expected sum of discounted cost under the task policy: $v^C = \mathbb{E}_{\pi_{\text{task}}} \left[ \sum_{t=1}^{\infty} \gamma^{t-1} \cdot c_t | s_0 = s \right]$. The critic is then incorporated into the task policy gradient as follows:

$$\nabla \mathcal{J} = \mathbb{E}_{\pi_{\text{task}}} \left[ \sum_t^H (G_t - \alpha G_t^C) \cdot \nabla \log \pi_{\text{task}}(a_t | s_t) \right], \tag{3}$$

where $G_t$ and $G_t^C$ are the reward and cost returns respectively. The hyperparameter $\alpha$ weights the impact $G_t^C$ has on $G_t$.

### 3.5 Style Shielding

The shielding procedure requires hyperparameters $\Delta$, called the safety level by Goodall & Belardinelli (2023), the approximation error $\epsilon$, the imagination horizon $H$, the incurred cost of a violation $C$ and a style specification $\Psi$. When the agent interacts with the environment, AMBS tries to prevent the task policy $\pi_{\text{task}}$ from violating the style specification $\Psi$. The shielding procedure works by generating $m$ trajectories $(\hat{s}_{1:H}, \hat{c}_{1:H}, \hat{\gamma}_{1:H})$ obtained by sampling the World Model $p_\theta$ with actions from $\pi_{\text{task}}$, where the estimated state is $\hat{s} = (z, h)$. We estimate the total cost of each trajectory as:

$$\text{cost}(\tau) = \sum_{t=1}^H (\hat{\gamma}_t)^{t-1} \cdot \hat{c}_t, \tag{4}$$

and verify whether it satisfies the inequality:

$$\text{cost}(\tau) < \gamma^{T-1} \cdot C. \tag{5}$$

Equation 5 signifies if a trajectory violates the style specification. Calculating the proportion of $m$ trajectories that satisfy the inequality gives a statistical estimate $\hat{\mu}_{s_t \models \Psi}$ that can be interpreted as the confidence that $\pi_{\text{task}}$ will not violate the specification. The estimate is calculated as a mean:

$$\hat{\mu}_{s_t \models \Psi} = \frac{1}{m} \sum_{i=1}^m \mathbb{1} \left[ \text{cost}(\tau_i) < \gamma^{T-1} \cdot C \right], \tag{6}$$

where $\mathbb{1}(\cdot)$ is the indicator function. If $\hat{\mu}_{s_t \models \Psi}$ is within the interval $[1 - \Delta + \epsilon, 1]$, the action from the task policy is deemed to follow the style specification and sampled from $\pi_{\text{task}}$, otherwise the action is sampled from the style policy $\pi_{\text{style}}$. To improve the predictive ability of the shielding procedure beyond the imagination horizon $H$, Goodall & Belardinelli (2023) propose to incorporate the critics from PENL into the trajectory cost as:

$$\text{cost}(\tau) = \sum_{t=1}^H (\hat{\gamma}_t)^{t-1} \cdot \hat{c}_t + \min(v_1^C(h_H, \hat{z}_H), v_2^C(h_H, \hat{z}_H)), \tag{7}$$

since a learned critic is able to evaluate the associated cost of a policy from a given state. AMBS allows shielding to be done even without complete knowledge of the environment dynamics. By learning a model and using it to predict how costs will accumulate with a given policy, it is possible to say how well the policy adheres to the style specification.

## 4 Experimental Setup

### 4.1 Safety Gym

Safety Gym is an environment introduced by Ray et al. (2019) to research constrained and safe deep reinforcement learning, and allows for several configurations of goals, environment objects, and agents. Figure 1 shows screenshots of the environment and agent. We configure Safety Gym to have the environment objective of reaching the goal area, seen as a green cylinder, while avoiding hazardous zones, marked as blue circles which is a typical objective in many games. In each episode, the position of game objects and the agent are randomly placed. The position of the goal area moves if the agent manages to reach it. The reward function of the environment is:

$$R = \begin{cases} d_t - d_{t-1}, s_t \neq \text{goal-area} \\ R_{\text{Goal}}, s_t = \text{goal-area} \end{cases} \tag{8}$$

where $d_t$ is the distance to the goal, and $R_{\text{Goal}}$ is a scalar constant reward for reaching the goal. The associated cost with being in a hazardous area is:

$$c_t = \mathbb{1}\left[\texttt{player-in-hazard}\right]. \tag{9}$$

The *Point* agent, seen in Figure 1, is a configurable agent of Safety Gym, and has access to two concurrent actions $A = (a^1, a^2)$, where $a^1$ controls forward and backward acceleration, and $a^2$ controls clockwise and counter-clockwise rotation. Both actions are continuous and in $a^1, a^2 \in [-1, 1]$. Movement is constrained to the 2D plane. Agent observations are made out of an aggregate of vector observations: agent position ($\mathbb{R}^3$) agent acceleration ($\mathbb{R}^3$), agent velocity ($\mathbb{R}^3$), agent angular velocity ($\mathbb{R}^3$), agent orientation relative to a true north direction ($\mathbb{R}^3$), and a pseudo-lidar. The pseudo-lidar works by iterating through all objects of interest (goal, hazardous areas) in the scene and if they are within range, the observation value is calculated as $O_i^{\text{lidar}} = D_i/D_{max}$, where $O_i^{\text{lidar}}$ is the $i$th lidar observation, and $D_i$ is the distance between an object of interest and the $i$th lidar, while $D_{\text{max}}$ is the maximum detection distance. There are 16 "rays" for the goal, and 16 "rays" for the hazardous areas ($\mathbb{R}^{32}$). The total observation space of the agent is a vector of size 47.

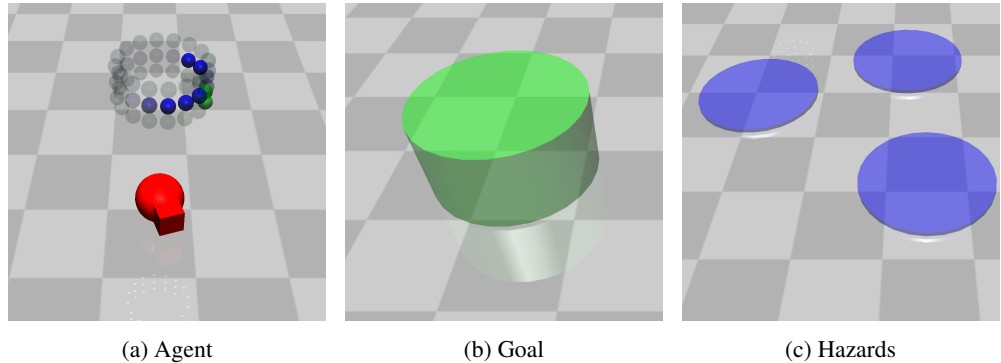

|            (a) Agent            |            (b) Goal            |            (c) Hazards            |

Figure 1: Images overlooking the Safety Gym environment. The point agent in red, goal area in green and hazardous zones in blue. Above the point agent an visualisation of the pseudo-lidar is present.

### 4.2 Style Specifications

In our evaluation, we define and utilise two style specifications $\Psi_1$ and $\Psi_2$ to explore how shielding can be used to control a RL-based NPC. Both specifications incorporate $q =$ `player-in-hazard` for the agent to learn to avoiding hazards, which can be inherently detrimental to episodic reward return. The first specification, called the *boundary* specification, applies

a shield that prevents the agent from being within a circle of radius 0.5 in the middle of the environment, even though the goal area can appear within the boundary. The specification is written as:

$$\Psi_1 = \texttt{outside-boundary} \wedge \neg\texttt{player-in-hazard}, \tag{10}$$

and shows how shields can be used to change states visited by an NPC. The second specification, called the *goal-heading* specification, applies a shield which prohibits the agent from moving towards the goal backwards. The specification is implemented by labelling states where the agent deviates by more than $5°$ from the vector pointing towards the goal as a violation. The goal-heading specification is written as:

$$\Psi_2 = \texttt{goal-headed} \wedge \neg\texttt{player-in-hazard}. \tag{11}$$

The goal-heading specification influences the agent to act more "natural" by preferring to move forwards rather than backwards. This mitigates a type of issue that can commonly occur with RL due to an agent reward hacking, since the agent does not care about acting naturally as long as reward is collected.

### 4.3 Training Setup

AMBS was implemented with Jax. For each shield scenario, the agents were trained for a 1000000 steps with 1000 steps per episode. For training, 10 parallel environments were used on an NVIDIA RTX 4090 with 24GB VRAM, an Intel 13th Gen Core i9-13900K, and 64GB of RAM.

## 5 Results

Our main goal is to explore how shielding, specifically AMBS, can be used in a game development setting to control a task-oriented RL policy. The agent should follow the predefined style specification, but also try to maximise environment reward, and balance task and style objectives. Evaluation was done in the Safety Gym environment with two different style specifications, which are detailed in Section 4.

### 5.1 Boundary Specification

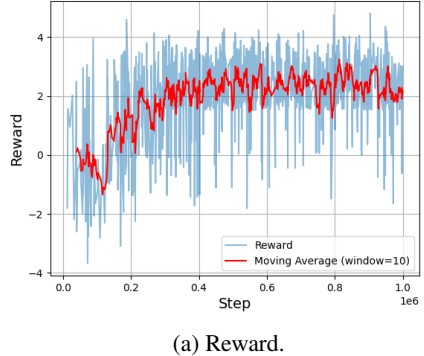

(a) Reward.

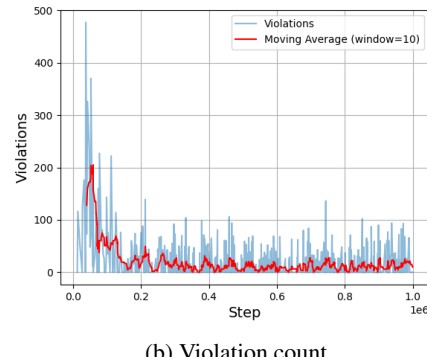

(b) Violation count.

Figure 2: Episodic reward and violation count for the boundary agent during 1000000 steps of training.

The episodic reward and violation count for the boundary agent can be seen in Figure 2. The agent is managing to increase the episodic reward of the environment while also minimising the total number of style violations of moving within the boundary and hazardous zones. We also observed that the agent could fail to reach the goal by becoming stuck trying to traverse the boundary area to get to the goal, but was stopped by the shield and stayed in place.

## 5.2 Goal-Heading Specification

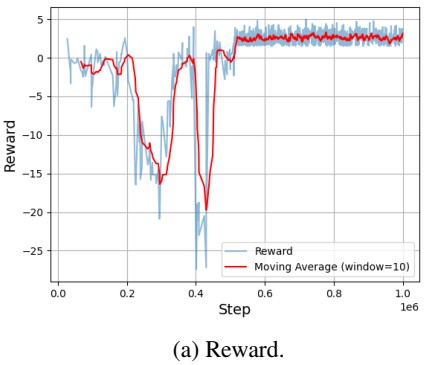

(a) Reward.

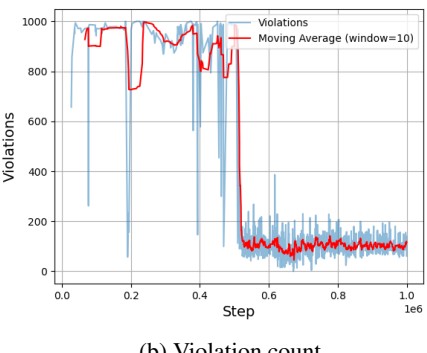

(b) Violation count.

Figure 3: Episodic return and violation count for the goal-heading agent during 1000000 steps of training.

The episodic reward and violation count for the goal-heading agent can be seen in Figure 3. The episodic reward return increases with training, while also minimising the associated cost. Also, by observing the policy in the environment, we could observe that the agent learned the behaviour of rotating its body towards the goal and then start moving towards it.

## 6 Conclusion and Discussion

We explore how Approximate Model-based Shielding can be used to design NPC behaviour. Preliminary results suggest that AMBS can use predefined style specifications to monitor and augment an RL agent's actions. We evaluated how an agent is able to both follow the environment objective of reaching a goal area and avoiding hazardous zones while also following the style specifications: boundary, goal-heading, enforced with shielding. The results indicate that shielding could be used to align RL policies with sought after gameplay behaviour specified by game developers. Thereby allowing them to combine their domain expertise while also benefiting from the complex behaviours RL can learn, something not possible with classical techniques.

As an exploration of shields for NPC behaviour design, there are limitations to the scope of this work. Firstly, shielding for NPC behaviour was evaluated within Safety Gym, and while the state and action spaces are continuous and environment mechanics are akin to gameplay, it is does not capture all aspects of modern video games. Also, the style specifications defined for the point agent are simple. NPCs can be expected to show complex and nuanced behaviours, and players are perceptive and have preferences regarding how an NPC acts. Shields are created in a quite different manner from conventional NPC behaviour designing techniques. Having to express and combine atomic propositions, such as never be out-of-bounds $\phi : \neg\texttt{out-of-bounds}$ and train an RL policy around it, might not be easy to incorporate into for game developer workflows. For future work, we will address the limitations of this paper. Firstly, to test shielding in a modern video game setting with complex game mechanics, higher-dimensional state and action spaces, and multiple agent, as has been done with other RL techniques (Vinyals et al., 2019; Berner et al., 2019; Vasco et al., 2024; Hafner et al., 2025; Wurman et al., 2022). Secondly, we would like to define style specifications and shields to express complex, believable and human-like NPC behaviour, and also evaluate how human players perceive them. Lastly, it would be interesting to explore if style specifications could be generated with modalities, such as natural language, since it has been done for specification languages in other domains (Wang et al., 2021; Liu et al., 2022; Chen et al., 2023), and also study how it can impact the workflow of game developers.

**Acknowledgments**

This work was partially supported by the Wallenberg AI, Autonomous Systems and Software Program (WASP) funded by the Knut and Alice Wallenberg Foundation.

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
