# OpenReview forum: "Exploring Model-based Shielding for Non-Player Character Behaviour"
_rl-conference.cc/RLC/2025/Workshop/RLVG — RLVG Workshop - RLC 2025_

### Official Review · Reviewer_91FA · 2025-06-16
**Review for Workshop RLVG Submission18**

**Rating:** 3
**Confidence:** 3

**Summary:**

The study shows how the shielding technique can be utilised to specify an NPC behaviour. The authors repurpose shielding for RL to design NPC behaviours by using a world model for planning and swapping the actions which would break the given behaviour rules.

**Strengths:**

- The paper provides concrete examples of the usecases, benefits and potential of the proposed methodology
- The study uses an established method and repurposes it for an important aspect of NPC behaviour
- The methodology is separate from the reward function maximisation, allowing for direct modification of the NPC behaviour after training
- The experiments support the claims of the authors

**Weaknesses:**

While the technical aspect of the work seems strong, my main concern about the method is the idea that an NPC's behaviour needs to be designed by restricting specific behaviours.  If the goal is for the game designer to create behaviours, it seems counterintuitive to generate behaviours by restricting atomic elements rather than promoting them. It would be interesting to know how this approach would scale to a more complex behaviour, from simple movement restrictions to creating an NPC that is a "chatty person"..

If a specific behaviour for a risk-averse agent is "only take the safest route", that would require many rules, and some dynamic adjustments, which are not easily representable by atomic formulas. It seems to me that the number of rules for shielding would grow in order to promote the desired behaviour, requiring many rollouts of the world model in order to satisfy the requirement. For the use of the NPCs in a real-time video game, this might become prohibitively expensive.

**Best Paper Nomination:**

No

**Claims:**

The authors' claims are supported by the experiments.

**Suggestions:**

See weaknesses

---

### Official Review · Reviewer_dMYA · 2025-06-17
**I'm leaning towards the acceptance of the paper as it seems to explore an interesting research direction. The authors provide preliminary experimental results supporting their proposed method; however, the experiments could be richer.**

**Rating:** 3
**Confidence:** 3

**Summary:**

The paper explores how approximate model-based shielding (AMBS) can be used to modulate RL agents' behaviours in the context of games. The authors provide preliminary results suggesting that AMBS can shape an agent's way of acting while still aiming for reward-maximizing behaviour.

**Strengths:**

- The paper seems to explore an interesting research direction.
- The idea seems novel (however, I'm not very familiar with previous works).
- Writing could be made clearer here and there, but overall, the paper is readable and is well-organised.

**Weaknesses:**

- The experiments could be more rich. In particular, there is no evidence in the document to support the discussion in Sec. 5.2 (could be nice to qualitatively study the behaviours). Also, no std, C.I.'s are displayed in the plots.
- The work should better connect with other tasks/frameworks such as imitation learning or constrained MDPs.

**Best Paper Nomination:**

No

**Claims:**

The authors propose to apply AMBS to video games. The authors provide preliminary results showcasing their method; however, the experiments/analysis could be richer.

**Suggestions:**

Maybe the authors could further elaborate on the connections with imitation learning and constrained MDPs (CMDPs). Regarding CMDPs, I see later around line 126 a discussion about them, but I suggest talking about them maybe earlier in the document.

I got a bit confused at some point regarding whether shielding is about the states the agents wants to visit, or the states that de agent does not want to visit. I know both are equivalent, but maybe I suggest the authors to emphasize this in the document. I got particularly confused around lines 115-121. I guess shielding is about states the agent does *not* want to visit, hence the negation.

Line 141, double comma.

Remove line 165.

What is $C$? It appears first around line 173, but I can't find its definition. Please clarify or remind the reader about it, particularly around line 184.

I was not able to follow the discussion well in lines 186-189. Please clarify.

Line 196, double "to".

Line 232, double ".".

I did not quite follow the discussion in lines 244-248.

---

### Decision · Program_Chairs · 2025-06-19

**Decision:**

Accept

**Comment:**

This paper investigates the application of approximate model-based shielding (AMBS) to modulate reinforcement learning (RL) agents' behaviors in video games, aiming to shape an agent's actions while maintaining reward-maximizing behavior.

The paper's strengths include its exploration of an interesting and novel research direction, providing concrete examples of the methodology's benefits and its potential to modify NPC behavior independently of the reward function.

However, the reviewers felt that the experiments could be richer, and the qualitative analysis and statistical measures could be improved. Additionally, the paper could better connect with related frameworks like imitation learning and constrained MDPs, while also clarifying certain confusing aspects of the shielding mechanism. We encourage the authors to address these points in the camera-ready version for presentation at the workshop.